Hepatic bile acids and bile acid-related gene expression in pregnant and lactating rats

Zhu Qiong N. 1
Xie Hong M. 2
Zhang Dan 1
Liu Jie 1 3 JLiu@kumc.edu
Lu Yuan F. 1 Yflu@zmc.edu.cn
1 Department of Pharmacology and Key Lab of Basic Pharmacology of Guizhou, Zunyi Medical College , Zunyi , China
2 Department of Gynaecology and Obstetrics, The Third Affiliated Hospital of Zunyi Medical College , Zunyi , China
3 University of Kansas Medical Center , Kansas City, KS , USA
Berdeaux Rebecca
Electronic publication date: 2013 Aug 27
Publication date: 2013
Volume: 1
Electronic Location ID: e143
Received 2013 Jun 24; Accepted 2013 Aug 5
Copyright: © 2013 Zhu et al.
Copyright year: 2013
Copyright holder: Zhu et al.
License: This is an open access article distributed under the terms of the Creative Commons Attribution License, which permits unrestricted use, distribution, and reproduction in any medium, provided the original author and source are credited.
License URL: https://creativecommons.org/licenses/by/3.0/

Keywords: Liver bile acids, FXR-SHP, Ntcp and Bsep, Cyp7a1, Pregnant and lactating rats, Oatps

Funding: Science and Technology Foundation of Guizhou Province QK2008-002 2009-70019 Foundation of Zunyi Medical College QK023 This study was supported by the Science and Technology Foundation of Guizhou Province (QK2008-002 and 2009-70019) and the Foundation of Zunyi Medical College (QK023). The funders had no role in study design, data collection and analysis, decision to publish, or preparation of the manuscript.

==============================
Background. Significant physiological changes occur during pregnancy and lactation. Intrahepatic cholestasis of pregnancy (ICP) is a liver disease closely related to disruption of bile acid homeostasis. The objective of this study was to examine the regulation of bile acid synthesis and transport in normal pregnant and lactating rats.

Materials and Methods. Livers from timed pregnant SD rats were collected on gestational days (GD) 10, 14 and 19, and postnatal days (PND) 1, 7, 14 and 21. Total bile acids were determined by the enzymatic method, total RNA was isolated and subjected to real time RT-PCR analysis. Liver protein was extracted for western-blot analysis.

Results. Under physiological conditions hepatic bile acids were not elevated during pregnancy but increased during lactation in rats. Bile acid synthesis rate-limiting enzyme Cyp7a1 was unchanged on gestational days, but increased on PND14 and 21 at mRNA and protein levels. Expression of Cyp8b1, Cyp27a1 and Cyp7b1 was also higher during lactation. The mRNA levels of small heterodimer partner (SHP) and protein levels of farnesoid X receptor (FXR) were increased during pregnancy and lactation. Bile acid transporters Ntcp, Bsep, Mrp3 and Mrp4 were lower at gestation, but increased during lactation. Hepatic Oatp transporters were decreased during pregnancy and lactation.

Conclusion. Hepatic bile acid homeostasis is maintained during normal pregnancy in rats, probably through the FXR-SHP regulation. The expression of bile acid synthesis genes and liver bile acid accumulation were increased during lactation, together with increased expression of bile acid efflux transporter Bsep, Mrp3 and Mrp4.

Introduction

Significant physiological changes occur during pregnancy and lactation to support the nutritional demands of the developing fetus and lactating pups (Carlin & Alfirevic, 2008; Athippozhy et al., 2011). Bile acids and cholesterol metabolism are important changes during pregnancy and lactation to support and to protect offspring development (Wooton-Kee, Cohen & Vore, 2008; Athippozhy et al., 2011; Abu-Hayyeh, Papacleovoulou & Williamson, 2013). Such physiological changes would also affect hepatic drug processing genes of phase-1, phase-2 metabolism and transporters (Aleksunes et al., 2012; Shuster et al., 2013). The alteration of bile acid homeostasis during pregnancy could unmask cholestatic disease in genetically predisposed but otherwise asymptomatic individuals (Milona et al., 2010). Recent work suggests that in pregnant mice farnesoid X receptor (FXR)-SHP (small heterodimer partner, NR0B2) regulation could be dysfunctional in its ability to downregulate the rate-limiting bile acid synthetic enzyme Cyp7a1 and 8b1, resulting in bile acids accumulation in the liver of late pregnancy mice (Milona et al., 2010; Aleksunes et al., 2012).

Intrahepatic cholestasis of pregnancy (ICP) is a liver disease which can occur in the third trimester of pregnancy (Abu-Hayyeh, Papacleovoulou & Williamson, 2013). The etiology and pathogenesis of ICP are still not clear, but many studies have related this disease to abnormal bile acid metabolism (Abu-Hayyeh, Papacleovoulou & Williamson, 2013; Floreani et al., 2013). ICP with elevated bile acids in serum and liver is a major cause for premature embryo development and embryonic death (Diken, Usta & Nassar, 2013). Genetic variations or mutations of farnesoid X receptor (FXR) (Van Mil et al., 2007), bile salt export pump (BSEP/ABCB11) (Dixon et al., 2009), and ATP-binding cassette, sub-family B (MDR/TAP), member 4 (ABCB4/MDR3) and ABCB11 (Dixon et al., 2000; Anzivino et al., 2013) contribute to the etiology of ICP. To fully understand bile acid synthesis, transport, and regulation in normal pregnancy would help us to shed light on the pathology of ICP.

Estradiol and/or its metabolites may interfere with FXR activity during pregnancy (Milona et al., 2010; Aleksunes et al., 2012), and a defect in progesterone metabolism is also implicated in the etiology of ICP (Pascual et al., 2002). Estrogen signaling is associated with pregnancy-induced hepatotoxicity and cholestasis in mice (Arrese et al., 2008), and reduced hepatic PPAR-α function in the mouse also appears to be estrogen-dependent (Papacleovoulou, Abu-Hayyeh & Williamson, 2011).

The above scenario has been studied extensively in mice (Milona et al., 2010; Aleksunes et al., 2012; Shuster et al., 2013). Mice and rats are the two most commonly used experimental animals, but some physiological responses are different. For example, in mice, Cyp7a1 and liver bile acid pool were not increased during lactation (Aleksunes et al., 2012), whereas the bile acid synthesis gene Cyp7a1 and hepatic bile acids are increased 2 to 3-fold in lactating rats (Wooton-Kee, Cohen & Vore, 2008; Wooton-Kee et al., 2010). In mice, pregnancy and lactation are associated with decreases in hepatic transporters, including bile acid transporters (Aleksunes et al., 2012), and such a phenomenon should also be characterized in rats. To fully understand bile acid synthesis, transport, and regulation in normal pregnancy would help us to shed light on the pathology of ICP. This study was initiated to investigate bile acid metabolism and transport gene expressions in pregnant and lactating rats, and the results suggest that under physiological conditions, FXR-SHP regulation might play a role in bile acid homeostasis in pregnant and lactating rats.

Materials and Methods

Animals

Adult Sprague Dawley (SD) rats (250 g) were purchased from the Experimental Animal Center of Third Military Medical College (Chongqing, China; certificate No. CXK 2007-0005). Rats were kept in a SPF-grade animal facilities (certificate No. SYXK 2011-004) at Zunyi Medical College, with regulated environment (22 ± 1°C, 50 ± 2% humidity and a 12 h:12 h light:dark cycle) and free access to purified water and standard rodent chow. Rats were acclimatized for 1 week, and subjected to timely mating overnight. A vaginal plug the next morning was designated as day 0 (GD 0) of gestation. Maternal livers were collected without fasting on GD10, GD14 and GD19, as well as on the postnatal days (PND) 1, 7, 14 and 21. The age-matched virgin rats were used as controls. The experimental design followed similar time points in mice (Aleksunes et al., 2012). Livers were weighed, snap frozen in liquid nitrogen, and stored at −80°C until analysis. All animal procedures followed the NIH guide of Humane Use and Care Animals, and were approved by Institutional Animal Use and Care Committee of Zunyi Medical College.

Bile acid determination

Bile acids were extracted from the liver and measured with the “Total” Bile Acid assay (TBA) kit (Nanjing Jian-Cheng Bioengineering Co., China). Briefly, livers were homogenized in physiological saline (1:9, wt:vol), followed by centrifugation at 2500 rpm/min for 10 min. The supernatant (30 µl) was taken for determination of bile acids according to the manufacturer’s protocol.

RNA Isolation and real-time RT-PCR analysis

Total RNA was isolated from frozen liver samples (50–100 mg) using 1 ml TRIzol (Takara, Biotechnology, Dalian, China) and subsequently purified with Total RNA (Mini) Kit (Watson Biotechnology, Shanghai, China). The quality of purified RNA was determined by spectrophotometry with the 260/280 ratio >1.8. Purified RNA was reverse transcribed with the High Capacity Reverse Transcriptase Kit (Applied Biosystems, Foster City, CA, USA). The primer pairs were designed with the Primer3 software and listed in Table S1. The Power SYBR Green Master Mix (Applied Biosystems, Foster City, CA, USA) was used for real-time RT-PCR analysis. The cycle time for reaching threshold (Ct) of each target gene was normalized to the housekeeping genes (G3PDH and β-actin), and expressed as % of housekeeping genes.

Western Blot Analysis

Livers were homogenized in a RIPA lysis buffer (Beyotime, P0013B, Shanghai, China) containing freshly-prepared proteinase inhibitors. The supernatants were centrifuged at 12000 rpm 10 min at 4°C, and protein concentrations were quantified by the BCA assay (Beyotime, P0012, Shanghai, China). Aliquot proteins were denatured with a protein loading buffer (Beyotime, P0015, Shanghai, China), and approximately 50 µg of protein/lane was separated on 10% SDS-PAGE and transferred to PVDF membranes. Membranes were blocked in 5% non-fat milk in TBST, followed by incubation overnight at 4°C with 1:1000 CYP7A1 (ab65586) and β-actin (Ab8227) from Abcam (Cambridge, MA), or FXR (sc-13063) from Santa Cruz Biotechnology (Santa Cruz, CA) in 1% BSA. After washing with TBST three times, the membranes were incubated with HRP-conjugated anti-rabbit or anti-mouse IgG (Beyotime, A0208 and A0216, Shanghai, China). Protein-antibody complexes were visualized using an enhanced chemiluminescent reagent (ECL-Plus) (Beyotime, P0018, Shanghai, China), and exposed to Gel Imaging (Bio-Rad, ChemiDoc XRS, USA). The intensity of the band was semi-quantified with Quantity One software.

Statistical Analysis

The software SPSS 17.0 was used for statistical analysis. The results were described using mean ± SEM. The difference between virgin and pregnant rats was determined by two-tailed independent samples test, P < 0.05 was considered statistically significant.

Results

Liver bile acid levels in pregnant and lactating rats

Bile acids were quantified in livers from control and pregnant rats at GD10, 14, and 19 and PND 1, 7, 14, and 21. Liver bile acid levels slightly decreased in late pregnancy, especially on GD 10, and 19. After birth, liver bile acid concentrations tended to increase, and there is a significant increase in PND 21 (30% over control) (Fig. 1).

Figure 1 Liver bile acid levels in pregnant and lactating rats.

Bile acids were quantified in livers from control and pregnant rats on GD10, 14, and 19 and PND 1, 7, 14, and 21. Dark gray bars represent pregnant rats, and black bars represent lactating rats. Data are presented as mean ± SEM (n = 3–6). Asterisks (*) represent statistically significant difference (p < 0.05) compared with the control.

Hepatic mRNA expression of bile acid synthesis genes in pregnant and lactating rats

The expression of the classic pathway bile acid synthetic enzyme genes (Cyp7a1 and 8b1) and alternative pathway (Cyp27a1 and 7b1), is shown in Fig. 2. The expression of rate-limiting Cyp7a1 mRNA was unchanged during pregnancy, and increased on postpartum. Cyp8b1 mRNA decreased in GD10 and GD14, and increased about 2-fold in PND14. The expression of alternative pathway genes Cyp27a1 and Cyp7b1 were unchanged in gestation days and increased in postnatal days.

Figure 2 Hepatic mRNA expression of bile acid synthetic pathway genes in pregnant and lactating rats.

The expression of bile acid synthetic classic pathway genes Cyp7a1, Cyp8b1 and alternative pathway genes Cyp27a1 and Cyp7b1 was quantified from control and GD10,14 and 19 and PND 1, 7, 14 and 21. Data were normalized to controls (set to 100%) and presented as mean ± SEM (n = 3–6). Dark gray bars represent pregnant rats, and black bars represent lactating rats. Asterisks (*) represent statistically significant difference (p < 0.05) compared with the control.

Hepatic expression of bile acid synthetic rate-limiting protein Cyp7A1 in pregnant and lactating rats

Western blots were performed using liver homogenates from control rats, pregnant rats at GD 10, 14, 19 and lactating rats at PND 1, 7, 14 and 21. The expressions of CYP7A1 protein were semi-quantified by band intensity. CYP7A1 protein was basically unchanged during pregnancy, a result similar to Cyp7a1 mRNA expression, but increased on lactation days PND7, 14 and 21 (Fig. 3).

Figure 3 Hepatic expression of bile acid synthesis rate-limiting protein CYP7A1 in pregnant and lactating rats.

Western blots were performed using liver homogenates from control, and pregnant rats in GD 10, 14, 19 and PND 1, 7, 14 and 21. The expression of CYP7A1 was semi-quantified by band intensity. Values are mean ± SEM (n = 5). Dark gray bars represent pregnant rats, and black bars represent lactating rats. The statistically significant difference was confirmed with a two-tailed independent samples test method (P < 0.05).

Figure 4 Hepatic mRNA expression of nuclear receptors SHP, FXR and ESR-1 and PPAR-α in pregnant and lactating rats.

The expression of bile acid regulation nuclear receptors genes SHP, FXR and Esr-1, PPAR-α were quantified using total hepatic RNA from control and pregnant mice at gestational days 10, 14, 19 and postnatal days 1, 7, 14, 21. Data were normalized to controls and presented as mean ± SEM (n = 3–6). Dark gray bars represent pregnant rats, and black bars represent lactating rats. Asterisks (*) represent statistically significant differences (p < 0.05) compared to the control.

Hepatic mRNA expression of nuclear receptors FXR, SHP, and ESR-1, PPAR-α in pregnant and lactating rats

The expression of bile acid regulation nuclear receptor genes farnesoid X receptor (FXR, NR1H4) did not show significant increases during pregnancy, while FXR gradually increased on postpartum. The small heterodimer partner (SHP; NR0B2) significantly increased in the late gestational days, increased 3-fold on GD 19 compared to controls. FXR plays an important role in bile acid homeostasis by inducing the transcription repressor SHP (Chiang, 2009). Estrogen receptor alpha (ESR-1) decreased to 64.7% and 57.7% on GD10 and GD14. In postnatal days, ESR-1 increased 2.33-fold in PND1 and then decreased to 68% of control on PND21. Proliferator-activated receptor α (PPARα) increased 3.79-fold compared to controls during lactation (Fig. 4).

Hepatic expression of FXR protein in pregnant and lactating rats

Western blots were performed using liver homogenates from control rats, pregnant rats at GD 10, 14, 19 and lactating rats at PND 1, 7, 14 and 21. The expression of FXR protein was semi-quantified by band intensity. FXR protein was increased during late pregnancy (GD10 to GD19) and early lactation (PND1 to PND7) (Fig. 5).

Figure 5 Hepatic expression of nuclear receptor FXR in pregnant and lactating rats.

Western blots were performed using liver homogenates from control, pregnant rats in GD 10, 14, 19 and PND 1, 7, 14 and 21. The expression of FXR was semi-quantified by band intensity. Values are mean ± SEM (n = 3). Dark gray bars represent pregnant rats, and black bars represent lactating rats. Statistically significant difference was confirmed with a two-tailed independent samples test method (P < 0.05).

Hepatic mRNA expression of bile acid transporters in pregnant and lactating rats

As illustrated in Fig. 6, the expression of bile acid efflux transporter bile salt export pump (Bsep/ABCB11) was decreased during pregnancy but increased during lactation. The multidrug resistance protein 3 (Mrp3) and Mrp4 showed a similar pattern, with slight increases during lactation. The ATP-binding cassette sub-family G member 2 (Abcg2/BCRP) was also decreased in the gestation and lactation days except PND1.

Figure 6 Hepatic mRNA expression of bile acid transporter in pregnant and lactating rats.

The expression of bile acid efflux transporter Bsep, Mrp3, Mrp4 and Abcg2 was quantified using total hepatic RNA from pregnant rats on GD 10, 14 and 19, and postpartum rats on PND 1, 7, 14 and 21. Data were normalized to controls and presented as mean ± SEM (n = 3–6). Dark gray bars represent pregnant rats, and black bars represent lactating rats. Asterisks (*) represent statistically significant differences (p < 0.05) compared to the control.

Hepatic mRNA expression of uptake OATP transporters and Ntcp in pregnant and lactating rats

Figure 7 demonstrates that the expression of canalicular uptake transporter solute carrier organic anion transporter (Oatp1/Slco1a1), solute carrier organic anion transporter (Oatp2/Slco1b2), and organic anion-transporting polypeptide 4 (Oatp4/Slc21a10) were all decreased in the gestation days, on PND1 Oatp1 increased 1.68-fold, then decreased in postnatal days. In comparison, Oatp2 and Oatp4 decreased in the both gestation days and lactation days. The uptake transporter Na+-taurocholate co-transporting polypeptide (Ntcp) was also decreased during pregnancy, increased on PND1, but decreased again thereafter during lactation.

Figure 7 Hepatic mRNA expression of uptake transporter Oatps and Ntcp in pregnant and lactating rats.

The expression of hepatic canalicular uptake transporter solute carrier organic anion transporters (Oatps) and Na+-taurocholate co-transporting polypeptide (Ntcp) transporters was quantified using total hepatic RNA from pregnant rats on GD10, 14 and 19, and postpartum rats on PND 1, 7, 14 and 21. Data were normalized to controls and presented as mean ± SEM (n = 3–6). Dark gray bars represent pregnant rats, and black bars represent lactating rats. Asterisks (*) represent statistically significant differences (p < 0.05) compared to the control.

Discussion

The present study demonstrates that in pregnant rats, hepatic bile acids were not elevated. Consistent with hepatic bile acid concentrations, bile acid synthesis genes and/or enzymes, i.e., Cyp7a1, Cyp8b1, Cyp27a1 and Cyp7b1 were not increased during pregnancy. Increased FXR protein and SHP mRNA are associated with bile acid homeostasis during pregnancy. In comparison, lactating rats had increased liver bile acid, increased bile acid synthetic enzymes, and increased expression of bile acid efflux transporters. In general, OATP transporters and bile acid uptake transport Ntcp were downregulated during pregnancy and lactation in rats.

ICP is characterized by raised serum bile acid levels and abnormal liver function tests (Geenes & Williamson, 2009; Diken, Usta & Nassar, 2013). However, in normal pregnant women, serum bile acid levels are not necessarily increased during pregnancy, regardless of gestation days (Barth et al., 2005; Egan et al., 2012). In experimental animal studies, a mild increase in liver bile acid levels during normal pregnancy in mice was reported in some studies (Aleksunes et al., 2012), but not in others (Abu-Hayyeh, Papacleovoulou & Williamson, 2013). In the majority of cases, such mild increases do not reach pathological levels and remain below the upper end of the reference range for serum bile acid levels (Abu-Hayyeh, Papacleovoulou & Williamson, 2013). Thus, it is not surprising that in the present study, liver bile acids were not elevated during pregnancy in gestation days (Fig. 1). The expression of bile acid synthesis gene and proteins during the gestation days (Figs. 2 and 3) is in agreement with hepatic bile acid profiles.

ICP has a complex etiology including genetic factors, endocrine factors, and the impact of pregnancy on FXR function (Abu-Hayyeh, Papacleovoulou & Williamson, 2013; Floreani et al., 2013). The present study focused on FXR-SHP regulation under physiological conditions. It is proposed that pregnancy in mice resembles a state of FXR inactivation (Milona et al., 2010; Aleksunes et al., 2012), and attenuated FXR function during mouse pregnancy has been reported (Papacleovoulou, Abu-Hayyeh & Williamson, 2011; Aleksunes et al., 2012) and the 3β-sulfated progesterone metabolite epiallopregnanolone sulfate was found to inhibit FXR, resulting in reduced FXR-mediated bile acid efflux (Abu-Hayyeh et al., 2013). In the present study, the expression of FXR mRNA in rats during pregnancy was basically unchanged. However, the FXR protein and FXR-inducible negative target SHP were markedly increased at the late gestation days and reached approximately 3-fold higher at GD14 and GD19, despite FXR mRNA not increasing. The increases in FXR-SHP may play an important role in maintaining the bile acid homeostasis and preventing the liver bile acids from accumulating to protect the fetus from the bile acid toxicity.

Lactation is a time of a five-fold increase in energy demand, as suckling young requires a proportional adjustment in the ability of the lactating dam to absorb nutrients (Cripps & Williams, 1975; Vernon et al., 2002). Lactating rats have a 2 to 3-fold increase in food consumption to ensure lactating dams absorb nutrients and synthesize critical molecules including bile acids to meet the dietary needs of the offspring and the dam (Vernon et al., 2002). The size and hydrophobicity of the bile acid pool increase during lactation, implying an increased absorption and disposition of lipid, sterols, nutrients, and xenobiotics (Athippozhy et al., 2011). In essence, rats (Wooton-Kee, Cohen & Vore, 2008) are different from mice (Aleksunes et al., 2012) in bile acid homeostasis during lactation. In the present study, hepatic bile acid pool (Fig. 1), mRNA levels of bile acid synthesis gene Cyp7a1, Cyp8b1, Cyp27a1 and Cyp7b1 (Figs. 2 and 3) were all increased during lactation, consistent with this scenario.

The mRNA levels of bile acid transporters Ntcp and Bsep followed a similar pattern. Ntcp is the major bile acid transporter for conjugated bile acid (Csanaky et al., 2011) and Bsep is the major bile acid efflux pump located at the bile canalicular apical domain of hepatocytes (Lam, Soroka & Boyer, 2010). Downregulation of Ntcp and Bsep was observed in pregnant rats (Arrese et al., 2003; Cao et al., 2001), however, they increase during early postpartum, probably under the influence of prolactin (Cao et al., 2001). In the present study, the changes in mRNA levels of Ntcp and Bsep showed a similar pattern, i.e., lower expression during pregnancy but returned to normal and even increased during lactation.

Sulfated progesterone metabolite (P4-S) levels are raised in normal pregnancy and elevated further in ICP, which can cause a competitive inhibition of NTCP-mediated uptake of taurocholate in Xenopus oocytes (Abu-Hayyeh et al., 2010), and also can cause inhibition of BSEP (Vallejo et al., 2006). In the present study, mRNA levels of Ntcp and Bsep (Fig. 5) were lower during pregnancy, and Bsep was increased during lactation, consistent with liver bile acid homeostasis profile. Mrp3 and Mrp4 are two major bile acid efflux (Cui et al., 2009; Aleksunes et al., 2012), and mRNA levels of these two genes expression showed the similar pattern (Fig. 5), i.e., lower during the pregnancy and higher during lactation. The pattern of these transporter mRNA levels coincide with FXR-SHP regulation of bile acid homeostasis, and fortifying the concept that under physiological conditions, FXR-SHP regulation of bile acid synthesis could be essential for maintaining the bile acid homeostasis and could prevent the occurrence of ICP, an unusual pathological condition.

One of the major findings in the study is the decreased mRNA levels of Oatp transporters (Fig. 6), and this finding is consistent with that observed in mice (Aleksunes et al., 2012; Shuster et al., 2013). Oatps are important not only for bile acid transport (Zhang et al., 2012), but also for drug and xenobiotic transport (Lu et al., 2008). In pregnant rats, the expression of Oatp2, but not Oatp1, was reported to decrease (Cao et al., 2001; Cao et al., 2002). The generalized downregulation of Oatp transporters could be an adaptive mechanism for the dam to protect developing fetuses and nursing pups from toxicants. Abcg2 is involved in epithelial transport/barrier functions, including bile acid transport (Blazquez et al., 2012). Abcg2 is proposed to play a key role in bile acid transport in placenta, as Bsep does in liver (Blazquez et al., 2012). In the present study, mRNA levels of Abcg 2 was depressed during pregnancy and lactation except for a transient increase at PND1. The pattern of Abcg2 expression is similar to Oatps, and can also be envisioned as an adaptive mechanism during pregnancy and lactation.

In summary, the present study suggests that in pregnant rats, FXR-SHP could regulate bile acid synthesis enzyme genes to prevent the accumulation of bile acids in the liver, together with downregulation of bile acid transporters Ntcp and Bsep. Pregnancy and lactation is associated with a general downregulation of Oatp and Abcg2 in rats. These data would add to our understanding of FXR-SHP regulation of bile acid homeostasis under physiological conditions.

Supplemental Information

Table S1 Supplementary Table 1

Click here for additional data file.

Additional Information and Declarations

Competing Interests

Author Contributions

Animal Ethics

Jie Liu is an Academic Editor for PeerJ.

Qiong N. Zhu and Hong M. Xie performed the experiments, analyzed the data, wrote the paper.

Dan Zhang performed the experiments, analyzed the data.

Jie Liu conceived and designed the experiments, performed the experiments, analyzed the data, wrote the paper.

Yuan F. Lu conceived and designed the experiments, analyzed the data, contributed reagents/materials/analysis tools, wrote the paper.

The following information was supplied relating to ethical approvals (i.e., approving body and any reference numbers):

All animal procedures follow the NIH guide of Humane Use and Care Animals, and were approved by Institutional Animal Use and Care Committee of Zunyi Medical College (2012-07).

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
