# Peer review of "Hepatic bile acids and bile acid-related gene expression in pregnant and lactating rats"

_PeerJ, doi:10.7717/peerj.143_

## Round 0.1 · original submission · Major Revisions

Two reviewers have evaluated the manuscript. While the study has merit, there are several issues to be addressed, some technical and many regarding the descriptions of the conclusions. Please respond to all reviewer comments. You may wish to consult with a language editing service. I agree with reviewer #1 that statements regarding the role of SHP are not supported by the data. The data show a possible correlation, but no loss of function studies are performed to demonstrate that SHP is in any responsible for the observed alterations in bile acids. The manuscript should be adjusted according to the comments of the reviewers.

One reviewer directs your attention to several previously published studies. Please consider how these may impact your work. You are not, however, required to cite any of these specific publications but may do so if you feel they are relevant. Inclusion of these specific citations will not be required for ultimate acceptance of a revision. I do, however, agree that more discussion and citations are warranted pertaining to the eitopathogenesis of ICP and the current knowledge about bile acid concentration in healthy pregnancy.

Reviewer 1 ·

Basic reporting

In this paper Zhu et al. have investigated the changes in the expression of genes involved in bile acid homeostasis during rat pregnancy and lactation. Although similar studies have been previously carried out the paper contain some interesting information mainly due to the fact that genes involved in metabolism and transport as well as key nuclear receptors, were included in the same study. Nevertheless, the manuscript contains important flaws that must be corrected before its publication could be recommended.

Experimental design

See General Comments for the Authors

Validity of the findings

See General Comments for the Authors

Additional comments

In this paper Zhu et al. have investigated the changes in the expression of genes involved in bile acid homeostasis during rat pregnancy and lactation. Although similar studies have been previously carried out the paper contain some interesting information mainly due to the fact that genes involved in metabolism and transport as well as key nuclear receptors, were included in the same study.

Main points

1. Throughout the text there is an important conceptual confusion among the terms homeostasis, metabolism, synthesis and transport. Homeostasis is maintained by several mechanisms, which include metabolism (synthesis and catabolism) and transport. Examples of misuse of terms:
Page 9: “regulation of homeostasis and transport”
Page 11: “”bile acid synthesis homeostasis”
Abstract: “closely related to bile acid metabolism” do you mean “homeostasis”?

2. ABSTRACT, Last sentence: “increased expression of bile acid transporters”. This is not true for all of them. I would suggest “changes in the expression of bile acid transporters” or “increased expression of major bile acid transporters”.

INTRODUCTION.
3. The introduction is focused on ICP, which does not match with the title or the actual study carried out. This is a basic descriptive study carried out under physiological conditions. Although a mention to the relevance of these data to further understand ICP the first paragraph should be shortened and placed later in the Introduction section.

4. If the etiopathogenesis of ICP is mentioned this should be done properly. The role of estrogens is mentioned (page 4) but not that of progesterone metabolites (Pascual et al. Clin Sci: PMID: 11980579).

5. In the second paragraph of page 4: “little is known the”. The sentence is wrong and not true.

METHODS

6. A limitation of the present work is the way total bile acids have been measured. Although in rat, which does not have gallbladder, an important part of bile acid pool is in the liver in fasting conditions, this is only part of the total pool.
a. The authors do not indicate whether the rats were fasted overnight.
b. Owing to their marked lipophilicity, the extraction of bile acids from liver tissue by centrifugation of homogenate diluted with saline is a very poor method.
c. In Figure 1: Liver bile acids are given as µmol/L. What does this mean? Surprisingly control value is 100.

DISCUSSION:
7. Page 10, first paragraph: I disagree that serum bile acid concentrations are not increased during healthy pregnancy. There are several reports in this sense. See for instance the paper already mentioned above (Pascual et al., Clin Sci: PMID: 11980579).

8. Page 12, first paragraph: There are important discrepancies between this study and that by Cao et al., 2001 that must be highlighted and commented.

9. Page 12, Second paragraph: Mechanism of progesterone metabolites-induced impairment in bile secretion includes both NTCP inhibition and, probably more importantly, the inhibition of BSEP (Vallejo et al., J. Hepatol. PMID: 16458994).

10. Page 12, Last of second paragraph: the sentence “to avoid…” is highly speculative and not supported by or even related to the present study.

11. Figures 5 and 6. It would make more sense to show all SLC transporters in one figure and all ABC pumps in the other., i.e., Exchange Ntcp with Abcg2. In addition, here and in the text use the updated nomenclature of rat Oatp transporters.


Minor Points
1. Abstract: “H epatic bile acid homeostasis maintained” correct to “Hepatic bile acid homeostasis is maintained”.

2. Introduction: Pag. 4, last line: “in pregnant rats” change to “pregnant and lactating rats”.

3. Page 8: “64.7%, 57.7% on GD10 and GD14” change to “64.7% and 57.7% on GD10 and GD14, respectively”.

4. Discussion: “SHP would be responsible”. This is not supported by the results and is probably wrong. I would suggest: “SHP may play an important role in”.

5. Figure 5: “Ntcp” label is to close to the Y-axis.

Reviewer 2 ·

Basic reporting

This work is very interesting and enhances my understanding of hepatic bile acids and bile acid-related gene expression in pregnant and lactating rat.

Experimental design

The authors researched bile acids synthesis by detecting bile acid synthesis enzymes, bile acids transporters and the regulation of bile acids homeostasis by detecting FXR-SHP gene expression in pregnant and lactation rat.

Validity of the findings

The study was initiated to investigate FXR-SHP regulation of bile acid homeostasis and transport in rats during pregnancy and lactation.

Additional comments

However, the authors should prepare a revised version of the paper, taking into account the following points
1) I am wondering why the authors collected the test time on gestation days 10,14,19 and postnatal days 1,7,14 and 21.
2) I am wondering whether the authors provide some protein expression proof for Cyp8b1, SHP and FXR.
3) I think this is a typo “PXR-SHP” on page four. It is FXR-SHP, is right?

---

## Round 0.2 · Minor Revisions

Thank you for your revision. Although some changes to the manuscript were made and new data presented, the manuscript does not yet meet publication standards. I had previously requested that statements about the role of FXR-SHP be strongly modulated or removed. These statements are still too strong, as no data presented show specifically that FXR-SHP regulate bile acid levels in your model. Please respond to the following specific points in a revised version.

Major points:

1. Page 5 “… results clearly demonstrate that under physiological conditions, FXR-SHP regulation plays important roles in bile acid homeostasis in pregnant and lactating rats.” This statement is not substantiated by the data. Page 13: The concluding sentence “The present study clearly demonstrates that in pregnant rats, FXR-SHP regulates bile acid synthesis enzyme genes to prevent the accumulation of bile acids in the liver, together with down-regulation of bile acid transporters…” is not substantiated by the data. An experiment showing that inhibition or activation of FXR-SHP causes alterations of bile acid synthesis genes in pregnant or lactating rats would be required. Statements about involvement of FXR-SHP should be substantially reduced and congruent with the level of analysis shown (correlation). At best, the data suggest a correlation.

Page 10: Please include a citation for the statement “Increased FXR protein and SHP mRNA play an important role in bile acid homeostasis during pregnancy.” If this is meant to refer to the present study, the sentence should be rewritten to only describe the correlation observed. No loss of function studies were presented to show that the alterations in mRNA or protein levels had an effect on bile acid homeostasis. Similarly, I do not agree that a correlation “adds to our understanding of FXR-SHP regulation of bile acid synthesis and transport in rats during pregnancy and lactation.” The statement “estrogen and FXR interactions may not be evident in rats…” is not supported by QPCR data. Please omit.

2. I cannot find anywhere where the n number of animals per condition is stated. Please state in the methods and the figure legends to go along with the discussion of statistical analysis. QPCR data should represent biological, not technical replicates and this should be clearly stated. For western blot data, single samples of each time point are shown; how were error bars generated and from how many additional experiments and replicates?

3. The western blots in Fig 3 appear to be from two different gels. Loading controls should be from the same gel. Why does the tubulin blot smile considerably but Cyp7A1 does not, although these are similar predicted molecular weight proteins (57 kDa/ 55 kDa)? If these two blots were from the same gel, both sets of bands should have the same shape.

4. In general, there seems to be an assumption that mRNA expression is reflective of protein abundance. For example, page 10 “Consistent with hepatic bile acid concentrations, bile acid synthesis enzymes… were not increased during pregnancy.” With the exception of CYP7A1, no protein data are shown for the other enzymes, so it is appropriate to specify that mRNA expression was not altered. As written, this sentence sounds like protein abundance was evaluated. Use of appropriate formatting for mRNA names will help make this more clear.

In the discussion, mRNA data are over-interpreted. The paragraph about BSEP, Mrp3 and Mrp4 sounds like the proteins are being discussed, when mRNA data was presented, which does not necessarily reflect the protein content. The discussion should be re-written to make these points clear to readers and to acknowledge the limitations of the study design.


Minor revisions:


The authors were suggested to consult an English language editing service. There remain numerous typographical errors and mistakes with English usage. As a courtesy, I have listed several corrections, but the authors are urged to take advantage of professional editing services.
ο Abstract, line 4 “ … pregnant and lactating rats”
ο Line 4: “timed pregnant”
ο Page 3 “asymptotic” should be “asymptomatic”
ο Page 3 “FXR … regulation mechanism “ delete “mechanism” … “for its ability” change to “in its ability”
ο Page 3: ICP is a liver disease that
ο mRNA and protein names should follow appropriate nomenclature throughout: mRNA- italics with the first letter capitalized “Cyp7a1”; protein- all caps. See MGI database website for assistance.
ο Page 9 “western bolts” should be “western blots”; “…the expressions of FXR protein…” should say “the expression”
ο Page 9 “..the expression of bile acid efflux… were decreased…” should be “was decreased”
ο Page 9 “with slightly increase during lactation” is incorrect “with slight increases during lactation”

Reviewer 1 ·

Basic reporting

No further comment

Experimental design

No further comment

Validity of the findings

No further comment

Additional comments

No further comment

---

## Round 0.3 · accepted · Accept

Thank you for the revision.